# Classification of Appearance Quality of Red Grape Based on Transfer Learning of Convolution Neural Network

**Zhihua Zha** [1]**, Dongyuan Shi** [2,3]**, Xiaohui Chen** [3]**, Hui Shi** [1] **and Jie Wu** [1,4,5,]*****

[1]  College of Mechanical and Electrical Engineering, Shihezi University, Shihezi 832003, China; zzh_inf@shzu.edu.cn (Z.Z.); 20212109003@shzu.edu.cn (H.S.)

[2]  Department of Horticulture, Agricultural College of Shihezi University, Shihezi 832003, China; ayeanchente@gmail.com

[3]  Research Center of Information Technology, Beijing Academy of Agriculture and Forestry Sciences/National Engineering Research Center for Information Technology in Agriculture/National Engineering Laboratory for Agri-Product Quality Traceability/Meteorological Service Center for Urban Agriculture, China Meteorological Administration-Ministry of Agriculture and Rural Affairs, Beijing 100097, China; chenxiaohui0326@163.com

[4]  Key Laboratory of Northwest Agricultural Equipment, Ministry of Agriculture and Rural Affairs, Shihezi 832003, China

[5]  Engineering Research Center for Production Mechanization of Oasis Characteristic Cash Crop, Ministry of Education, Shihezi 832003, China

*  Correspondence: wjshz@126.com

**Abstract:** Grapes are a globally popular fruit, with grape cultivation worldwide being second only to citrus. This article focuses on the low efficiency and accuracy of traditional manual grading of red grape external appearance and proposes a small-sample red grape external appearance grading model based on transfer learning with convolutional neural networks (CNNs). In the naturally growing vineyards, 195,120,135 samples of Grade I, Grade II, and Grade III red grapes were collected using a Canon EOS 550D camera, and a data set of 1800 samples was obtained using data enhancement technology. Then, the CNN transfer learning method was used to transfer the pre-trained AlexNet, VGG16, GoogleNet, InceptionV3, and ResNet50 network models on the ImageNet image dataset to the red grape image grading task. By comparing the classification performance of the CNN models of these five different network depths with fine-tuning, ResNet50 with a learning rate of 0.001 and a loop number of 10 was determined to be the best feature extractor for red grape images. Moreover, given the small number of red grape image samples in this study, different convolutional layer features output by the ResNet50 feature extractor were analyzed layer by layer to determine the effect of deep features extracted by each convolutional layer on Support Vector Machine (SVM) classification performance. This analysis helped to obtain a ResNet50 + SVM red grape external appearance grading model based on the optimal ResNet50 feature extraction strategy. Experimental data showed that the classification model constructed using the feature parameters extracted from the 10th node of the ResNet50 network achieved an accuracy rate of 95.08% for red grape grading. These research results provide a reference for the online grading of red grape clusters based on external appearance quality and have certain guiding significance for the quality and efficiency of grape industry circulation and production.

**Keywords:** grape; appearance quality; classification; convolutional neural network; transfer learning; support vector machine





## 1. Introduction

Red globe grapes, also known as red earth grapes, are a popular variety of fresh eating grapes in China due to their large size, vibrant color, and ability to be stored for extended periods of time. They are the second most widely cultivated variety of fresh eating grapes in the country [1]. Fruit appearance is the most intuitive quality characteristic of fruit and

accounts for 60% of the evaluation of fruit quality [2], directly affecting market prices and consumer purchasing desire. Red grape clusters exhibit inconsistency in size and irregular outlines. Furthermore, the grapes within the same cluster vary in size and color, posing challenges in achieving the desired grape grading outcome. Currently, the grading of red globe grapes mainly relies on subjective human classification. However, this method, which relies on the human eye to determine the grade of grapes, is highly subjective, prone to misjudgment, and leads to fatigue and inefficiency [3]. Therefore, there is an urgent need to achieve rapid and non-destructive grading of the appearance quality of red globe grapes to improve their commercial yield.

To address the aforementioned issues, domestic and foreign researchers have explored the use of machine vision technology. Chen Ying et al. [4] used three methods, namely the fruit surface coloring rate calculated in the RGB color space, the projection area method for calculating the size of grape clusters, and the projection curve method for calculating shape parameters of grape axes, to grade the appearance quality of 20 clusters of Kyoho grapes. The grading accuracy rates were 88.3%, 90.0%, and 88.3%, respectively. Li Junwei et al. [5] used machine vision technology to predict and grade the quality and diameter of individual grapes from seedless white and red globe grapes in Xinjiang, achieving a grading accuracy rate of over 85%. Xiao Zhuang et al. [6] used the random least-squares ellipse detection method to extract grape size and grade 42 clusters of red globe grapes, with a grading accuracy rate of 90.48%. The aforementioned studies all used single features as grape grading indicators, requiring image preprocessing and the selection of artificially extracted shallow features, which lacked robustness.

In recent years, deep learning has emerged as a modeling method based on computers [7]. Among them, convolutional neural networks are directly driven by data, enabling self-learning and avoiding the complex operation of manual feature extraction [8]. They also have good adaptability to image displacement, scaling, distortion, and the ability to combine low-level features into high-level features. As a result, they have been widely applied in crop identification [9,10], agricultural product quality detection [11–13], and crop disease identification fields [14,15]. Geng et al. [16] designed a dual-branch deep fusion convolutional neural network (DDFnet) for the classification of dried red dates. Li et al. [17] conducted comprehensive analysis based on the intact degree, fruit size, and color of green plum fruits, and developed a deep learning-based classification method for green plum grades with powerful feature extraction and recognition capabilities. They also established a cognitive error entropy based on the generalized entropy theory to reflect the credibility of the classification results. Momeny et al. [18] achieved recognition and grading of cherry fruit based on whether the fruit shape was regular by improving the traditional CNN model. Sozzi M et al. [19] compared three YOLO series algorithms for automatically detecting and counting white grape clusters, as an alternative to using object detection to estimate crop yields. Gulzar et al. [20] used a hybrid transfer learning approach to classify 40 fruit images by improving the MobileNetV2 network model architecture and used several different preprocessing model tuning techniques to prevent model overfitting with 99% recognition accuracy. Gulzar et al. [21] proposed a seed classification based on convolutional neural network and transfer learning models, which improves the model performance by attenuating the learning rate, model checkpointing, and hybrid weight adjustment, and the model achieves a classification accuracy of 99.9%. Mamat et al. [22] proposed an automatic image annotation method to annotate oil palm fresh fruit bunches with different maturity levels, which solves the problem of fast annotation of the dataset. Aggarwal et al. [23] demonstrated that combining different convolutional neural networks provides better prediction than a single model.

The deep feature learning and extraction of convolutional neural networks relies heavily on a large amount of data. In the case of a small sample size, overfitting can be a serious problem. Transfer learning reduces the amount of training data and computational power required to build deep learning models through knowledge (weight) sharing techniques, relaxing the conditions of the sample size and identical probability distribution, and can

effectively solve the problem of overfitting on complex network structures with small samples. Behera SK [24] compared classification methods using machine learning and transfer learning and ultimately achieved 100% accuracy in accurately classifying papaya ripeness using the VGG19 model based on transfer learning. Xue Yong et al. [25] used the GoogLeNet deep transfer model to detect apple defects. Initializing the network with transfer features can improve the network's generalization performance. However, the features extracted by each layer of the network model are different and have varying effects on classification performance. Yosinski et al. [26] discussed the feature extraction ability of different convolutional layers in the network structure under different data set sizes by freezing the parameters of each convolutional layer separately. The study showed that the classification performance of the transferred network is not necessarily increased with the increasing number of layers in the network.

In order to enhance the accuracy of grading the appearance quality of red grape clusters, this study employs the migration learning method. Five classical convolutional neural network models with varying network depths are compared and analyzed, and hyperparameter optimization is conducted to determine the optimal feature extractor. By combining SVM technology and analyzing the impact of feature outputs from each layer of the model on its performance, we achieve rapid and highly accurate grading of red grape clusters through optimal adaptation of the number of migrated layers. This is accomplished by integrating the migrated source model with the adaptive fine-tuning method, particularly in cases with limited datasets. The findings of this study provide valuable insights for online grading systems that focus on the appearance quality of red grape clusters.

## 2. Materials and Methods

### 2.1. Image Acquisition of Red Grape Ear

The red seedless grape clusters were harvested from the 22nd company of the 121st regiment of the 8th Agricultural Division of Xinjiang Production and Construction Corps, located at 44°81′ N, 85°59′ E and an altitude of 245 m. Professional personnel selected 65 first-grade clusters, 40 second-grade clusters, and 45 third-grade clusters based on the four appearance indicators of fruit powder integrity, cluster shape, berry diameter, and berry count, in accordance with the grading standard DB65/T2832-2007 [27] for red globe grapes. The grape clusters were then photographed from multiple angles (clockwise rotation of 120 degrees) using a Canon EOS 550D camera at a distance of 600 mm, with an image resolution of 2976 × 3968 pixels. Specific example images of the dataset are shown in Figure 1.

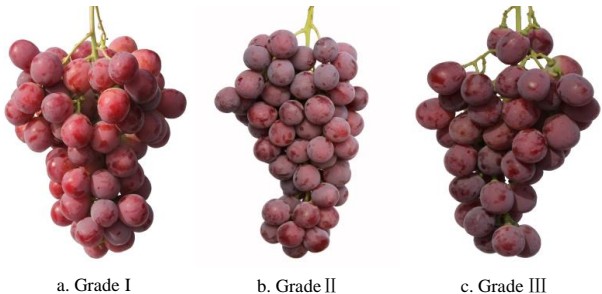

a. Grade Ⅰ          b. Grade Ⅱ          c. Grade Ⅲ

**Figure 1.** Example of "Red Globe" images data set of different grades.

### 2.2. Data Augmentation of Red Grape Cluster Images

Considering the randomness of grape cluster imaging and the necessity of data augmentation, the 450 images of the 150 grape clusters were randomly rotated by less than 15 degrees and Gaussian noise with mean 0 and variance 0.01, as well as salt and pepper noise with a noise density of 0.05, were added to increase the diversity of the training dataset, reduce overfitting during network training, and improve the generalization perfor-

mance of the network. Finally, the dataset of red grape cluster images was expanded to 1800 images. Subsequently, the grape cluster image dataset was randomly divided into a training set and a test set at a ratio of 7:3, with 1260 images used for learning the weight parameters of the grading model and 540 images used to verify the classification ability of the constructed model. The numbers of expanded red grape cluster images and dataset partitions are detailed in Table 1.

**Table 1.** Number of "Red Globe" sample images in each category.

| Cluster Grade | Original Image | Data Augmentation | Dataset Division | |
| --- | --- | --- | --- | --- |
| | | | Training Set | Test Set |
| Grade I | 195 | 585 | 546 | 234 |
| Grade II | 120 | 360 | 336 | 144 |
| Grade III | 135 | 405 | 378 | 162 |

*2.3. Construction Method of Appearance Quality Grading Model for Red Grapes*

2.3.1. Transfer Learning

As deep convolutional neural networks, represented by VGG16, ResNet50, and GoogleNet, have been sufficiently trained on the ImageNet dataset and learned a large amount of knowledge required for image classification recognition, this paper uses pre-trained classic deep convolutional neural networks as the basis and adopts the transfer learning method of pre-trained network weight parameters, which can reduce the effect of environmental factors on model performance. AlexNet, VGG16, ResNet50, GoogleNet, and InceptionV3 are used as feature extractors for red grape image classification recognition. Each convolutional neural network is composed of multiple convolutional and pooling layers. Among them, the convolutional layers load pre-trained weights and bias parameters to extract features and feature mapping from the input image, while the pooling layers are used to maintain the invariance of image features and obtain feature vectors for classification after the fully connected layer.

2.3.2. SVM Classifier

Support vector machine (SVM) is a model discriminant algorithm based on the idea of minimizing structural risk, which has superior classification performance compared to the end-classification function in traditional CNN, especially in solving small sample datasets, nonlinear classification problems, and high-dimensional pattern recognition, and can make better use of the knowledge learned by the pre-trained model [28]. Considering the small number of red grape cluster classification categories and images in this study, the Gaussian radial basis function (RBF) is selected as the kernel function of SVM to achieve the classification and recognition of red grape cluster grades.

2.3.3. Red Grape Appearance Quality Grading Model Construction Process

As CNN pre-trained networks are good at image feature extraction, and SVM is good at small sample dataset classification, this study combines the CNN optimal network as the feature extractor with SVM classifier to construct a red grape grading model based on the CNN-SVM hybrid model. The specific construction process is shown in Figure 2. First, the red grape images in the training set are input into the pre-trained networks of AlexNet, VGG16, GoogleNet, InceptionV3, and ResNet50 for image feature information extraction. In this process, due to the different matching degrees between each network model and the image input size, the red grape cluster image size is automatically adjusted by randomly flipping along the vertical axis and randomly translating no more than 30 pixels in the horizontal and vertical directions, to adapt to the requirements of different network models for image input size. Then, by comparing the classification accuracy of red grapes by the SoftMax classifiers of the 5 CNN networks, the best CNN feature extractor is selected, and the optimal weight parameters of the selected network are determined by adjusting

the learning rate and training times and other network hyperparameters. Finally, the red grape cluster image feature information extracted from different convolutional layers of the selected CNN network is input into SVM for red grape cluster grading, and model performance evaluation indicators are calculated to establish the best network depth of the feature extractor in the red grape grading model. The hardware device used in the model construction process is a SAMSUNG desktop computer, with an Intel(R) Xeon(R) CPU E5-2620@2.10 GHz processor, 16 GB memory, Windows 7 operating system, and Matlab 2018b software for image processing and network training.

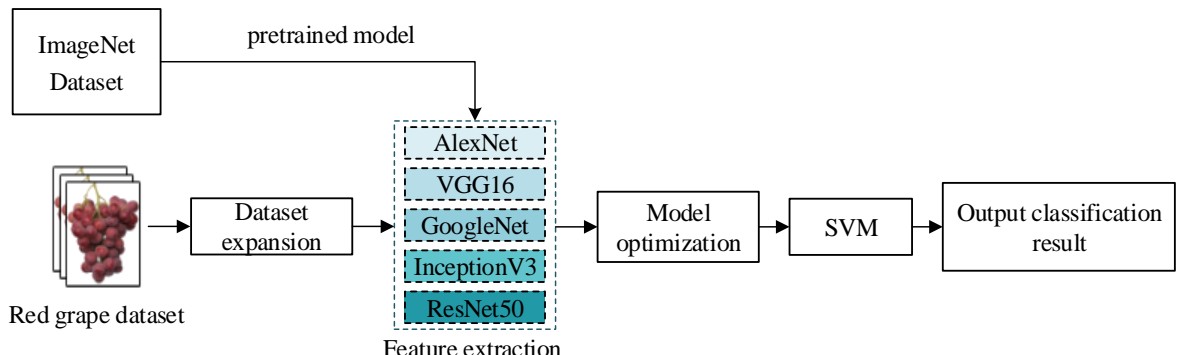

**Figure 2.** Flow chart of study.

### 2.4. Model Optimization

Since the sample size of the data set in this study is relatively small, in order to further improve the accuracy and real-time performance of image recognition of red grape ears, the established classification model needs to be optimized. This paper optimizes network performance by learning rate, training times, and optimizer. The optimizer adopts a stochastic gradient descent optimizer with a first-order momentum factor.

The classification categories in this paper are three grades of red grapes. However, in actual field environments, there tend to be fewer red grapes in the second class, making experimental samples more difficult to obtain. This tends to lead to the problems of unbalanced positive and negative samples and different levels of sample difficulty. For this reason, this paper used the weighted loss function to further improve the prediction ability of the model, and the loss function shown in Formula (1) is used to impose constraints and penalties on the model parameters through the L2 norm regularization term to prevent the network model from over-fitting and improve the generalization ability of the model.

$$\text{Loss} = -\frac{1}{N}\sum_{j=1}^{N}\sum_{i=1}^{N} y_{ji}\log(\hat{y}_{ji}) + \frac{\lambda}{2}\|W\|_2^2 \tag{1}$$

where $N$ represents the number of samples updated in one parameter update during model training, K represents the number of classification categories, $y_{ji}$ uses one-hot labels to represent the classification category labels for the $i$th image in the $j$th batch, $\hat{y}_{ji}$ represents the predicted classification category labels for the $i$th image in the $j$th batch, $\lambda$ represents the regularization parameter, and $W$ represents the weights.

### 2.5. Model Evaluation Metrics

Accuracy, recall, and F1-score were used to evaluate the classification performance of various network models for red grape bunches; the formulas are as follows:

$$\text{Accuracy} = \frac{TP + TN}{TP + TN + FP + FN} \times 100\% \tag{2}$$

$$\text{Recall} = \frac{TP}{TP + FN} \times 100\% \tag{3}$$

$$F_1 = \frac{2TP}{2TP + FN + FP} \times 100\% \tag{4}$$

where *TP* is the number of positive samples correctly determined as positive samples, and *TN* is the number of negative samples correctly determined as negative samples; *FP* is the number of negative samples wrongly determined as positive samples, and *FN* is the number of positive samples wrongly determined as negative samples.

## 3. Results and Analysis

### 3.1. Performance Analysis of Pretrained Network Models

Table 2 shows the classification results of different pretrained network models on red grape cluster images with the same hyperparameter settings, including learning rate (0.01), iterations (100), momentum factor (0.9), batch size (10), and dropout rate (0.5). As shown in Table 1, the detection speed of the AlexNet network model is the fastest, but its training and testing accuracy are relatively low, which may be due to the relatively shallow depth of the AlexNet network, leading to poor feature extraction ability of its convolutional layers for grape cluster images. With the increase in network layers, the classification performance of each model on red grape clusters is improved to varying degrees. Among them, the InceptionV3 network model has the best classification performance on the training set, while the ResNet50 network model has the highest classification performance on the testing set. Although the InceptionV3 and ResNet50 network models have their own strengths in the classification performance on the training and testing sets, respectively, it can be seen from the average detection time that the detection efficiency of the ResNet50 network model is much higher than that of the InceptionV3 network model, which can better meet the real-time requirements of online sorting in the future. Therefore, considering the classification performance of various network models on red grape clusters, this study uses ResNet50 as a feature extractor for deep exploration of red grape cluster images.

**Table 2.** Comparison of migration learning performance of different network models.

| Feature Extractor | Network Depth | Training Set | | | Test Set | | | Mean Detection Time/ms |
|---|---|---|---|---|---|---|---|---|
| | | Accuracy/% | Recall/% | F1-Score/% | Accuracy/% | Recall/% | F1-Score/% | |
| AlexNet | 8 | 75.62 | 75.86 | 74.97 | 72.31 | 70.09 | 72.35 | 97.00 |
| VGG19 | 16 | 81.35 | 82.24 | 80.81 | 78.15 | 75.46 | 77.67 | 128.64 |
| GoogleNet | 22 | 82.14 | 83.35 | 82.42 | 78.79 | 76.02 | 78.16 | 236.14 |
| InceptionV3 | 48 | 88.43 | 89.94 | 88.11 | 80.31 | 78.86 | 80.99 | 480.20 |
| ResNet50 | 50 | 86.74 | 87.43 | 85.98 | 82.85 | 80.31 | 82.69 | 241.20 |

### 3.2. Analysis of the Factors Affecting the Performance of the ResNet50 Model

During the training process, we used an exponential ruler to set the learning rate and conducted comparative experiments with different learning rates and iteration parameters to determine the optimal values of these parameters and further improve the transfer learning performance, using the average accuracy on the test set as the standard. According to the results in Table 3, under the same number of training iterations, when the learning rate is 0.001, the ResNet50 model performs better on the test set than other values in terms of average accuracy. As the number of training iterations increases, the average accuracy also increases to varying degrees. When the learning rate is 0.001, the average accuracy increases by 11.04%, 0.33%, and 0.13% as the number of training iterations increases. Moreover, when the number of training iterations reaches 40, the change in average accuracy is basically the same as 30 training iterations, indicating that the ResNet50 model stabilizes at 30 training iterations. After the number of training iterations exceeds 10, the improvement in the network's performance on the test set is limited. Considering the training time of the model, we ultimately set the learning rate to 0.001 and the number of training iterations to 10.

**Table 3.** Results of models training with different epochs and learning rates (%).

| Learning Rate | Epochs | | | | | |
|---|---|---|---|---|---|---|
| | 5 | 10 | 15 | 20 | 30 | 40 |
| 0.1 | 55.77% | 65.46% | 67.31% | 69.23% | 88.46% | 88.96% |
| 0.01 | 82.85% | 93.89% | 94.22% | 94.35% | 93.36% | 93.31% |
| 0.001 | 85.12% | 92.31% | 93.99% | 92.31% | 92.46% | 92.21% |

*3.3. Performance Analysis of Feature Extraction Based on ResNet50 Network Model*

The ResNet50 network model is divided into four stages, as shown in Figure 3. Each stage includes a residual mapping module and a varying number of identity mapping modules, enabling the network to become deeper while maintaining precision and controlling speed.

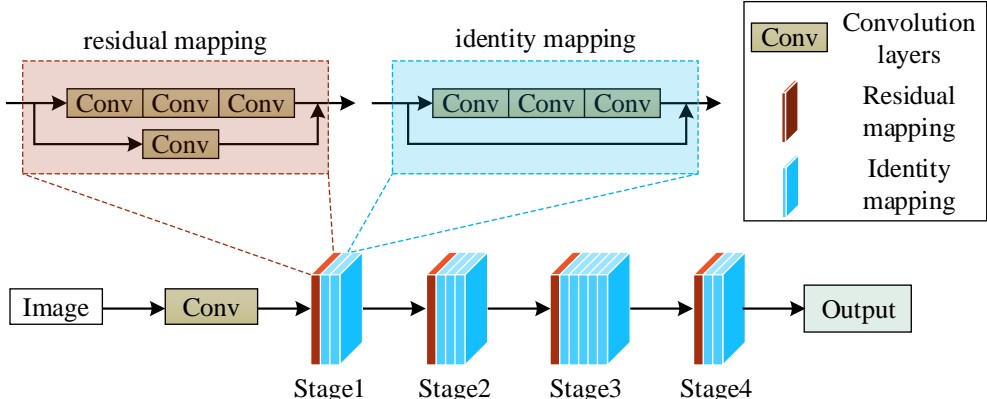

**Figure 3.** Structure of ResNet50 network.

3.3.1. Performance Analysis of ResNet50 Network Feature Extraction Based on Feature Visualization

Figure 4 shows the feature visualization of the four stages of the ResNet50 network. As can be seen from the figure, the shallow convolution layers near the network have smaller receptive fields for learning low-level features, such as color features, while the deeper convolution layers near the end of the network have larger receptive fields for learning higher-level combinations of low-level features to extract more advanced features.

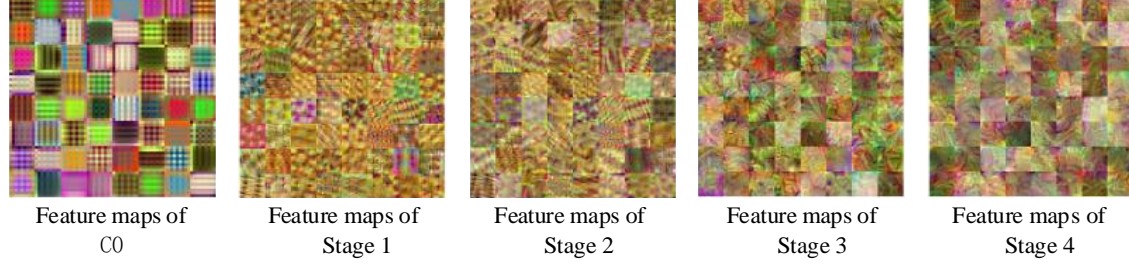

| Feature maps of C0 | Feature maps of Stage 1 | Feature maps of Stage 2 | Feature maps of Stage 3 | Feature maps of Stage 4 |

**Figure 4.** Example of feature extraction.

We selected the feature extracted by the 39th convolution kernel in the first convolution layer of ResNet50 and used it to activate the input image of a red grape. The resulting heatmap is shown in Figure 5b. The pixels closer to red indicate a stronger positive activation area. It can be seen from Figure 5b that this convolution kernel is a color filter that mainly extracts the feature of lost fruit powder on grape grains. For areas with strong light, it can better extract the region of lost fruit powder, while for areas with weak light, the feature extraction effect of lost fruit powder is poor.

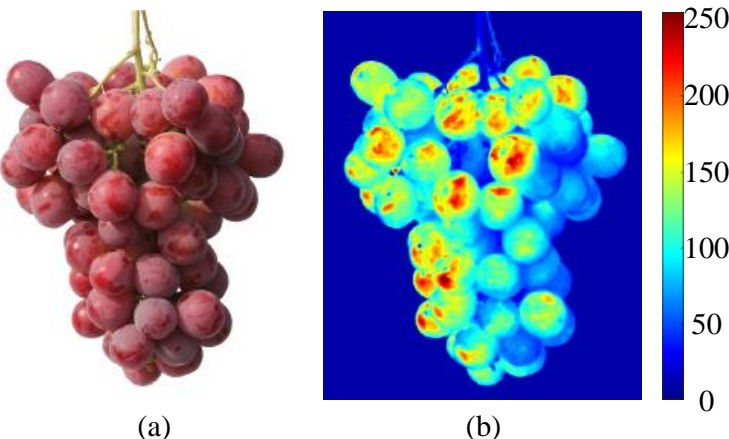

(a)        (b)

**Figure 5.** Example of feature extraction: (**a**) is the original image and (**b**) is the example of feature extraction.

### 3.3.2. Analysis of Network Architecture and Classification Performance of Red Grapes

To obtain better intuition of the features learnt by different module of the ResNet50, we cached the feature outputs of the convolution layers in the first convolution layer, each residual mapping module, and each identity mapping module (a total of 17 nodes) and trained a multiclass error-correcting output codes (ECOC) model using support vector machine (SVM) binary learners and a one-versus-one coding design. Using SVM to train the feature parameters of the 17 nodes extracted by the ResNet50 network model, the average accuracy of each node on the SVM is shown in Figure 6. With the increasing depth of the network, the average accuracy of the features extracted by each node on the SVM continued to increase. The highest average accuracy on the test set was achieved at node 10 (the first identity mapping module in Stage3), reaching 95.08%. At this point, the average accuracy on the training set was 96.88%. When the depth of the network continued to increase, the growth trend of the average accuracy on the training set slowed down, while the average accuracy on the test set showed a fluctuating downward trend. This is because the small size of the dataset and the complex deep network structure increased the likelihood of overfitting. The features extracted by nodes 1–10 are universal and can extract common feature information, while the features extracted by nodes 11–17 are specific to different datasets, so even if the network depth increases, the model's capability does not continue to increase. Therefore, we selected the output at node 10 of the ResNet50 network model as the feature for different levels of red grapes.

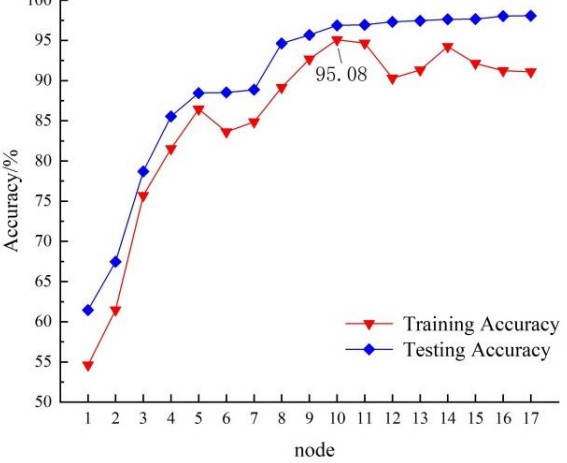

**Figure 6.** Classification accuracy of each node on test set.

Overall, this section analyzed the performance of the ResNet50 network model in terms of feature extraction and provided insights into the selection of appropriate features for different levels of red grapes.

### 3.3.3. Improved Model Performance Analysis

In this paper, the strongest feature extraction node in the output of the network model was selected as the feature extraction network, which was combined with SVM to build the model. From Table 4, the accuracy, recall, and F1-score of the ResNet50 + SVM method are 96.88%, 95.36%, and 96.43% at training, respectively. When the model is ResNet50 with softmax, the accuracy, recall, and F1-score are 86.74%, 87.43%, and 85.98%, respectively. Compared with the ResNet50 with softmax method, the accuracy, recall, and F1-score increase by 10.14%, 7.93%, and 10.45%, respectively. This indicates that the Resnet50 network model was combined with SVM to construct a grape quality grading model for red table grapes based on their appearance. At the same time, the accuracy, recall, and F1-score increase by 10.14%, 7.93%, and 10.45% at the test stage, respectively.

**Table 4.** Comparison of the performance of different grading methods.

| Grading Methods | Training Set | | | Test Set | | |
|---|---|---|---|---|---|---|
| | Accuracy/% | Recall/% | F1-Score/% | Accuracy/% | Recall/% | F1-Score/% |
| ResNet50 + softmax | 86.74 | 87.43 | 85.98 | 82.85 | 80.31 | 82.69 |
| ResNet50 + SVM | 96.88 | 95.36 | 96.43 | 95.08 | 94.88 | 96.78 |

Moreover, the appearance of 50 red table grape clusters were classified by this network model. Among them, 27 clusters of Grade I and Grade III were correctly classified according to the judgement of professional personnel, while three clusters of Grade II were misclassified as Grade III, resulting in an accuracy rate of 94%. Further analysis of the misclassified grape clusters revealed that in the three images taken of each cluster, there was a difference in shape or grade requirements on one side of the grape cluster. In the future, further research will be conducted on the comprehensive evaluation of grape cluster grades based on multiple images.

### 4. Conclusions

This article investigated a grape quality grading system network model based on transfer learning under small sample conditions. The main conclusions are as follows:

- By using a model-based transfer learning method, compared with five pre-trained network models, namely, Alexnet, VGG16, Googlenet, ResNet50, and InceptionV3, ResNet50 network model is more suitable for the red globe grape dataset with the same hyperparameter settings. On the test set, the average classification accuracy of the ResNet50 network model can reach 82.85%, and the F1 value is 82.69%.
- By optimizing the hyperparameters, when the learning rate is set to 0.001 and the number of iterations is 10, the grading accuracy of the ResNet50 network model on the red table grape dataset can reach 93.89%.
- By analyzing the feature output of the intermediate convolutional layers in the ResNet50 network model layer by layer, and combining SVM technology, a grape quality grading model for red table grape clusters based on their appearance was constructed and tested on the dataset. The average classification accuracy of the constructed network model can reach 95.08%. The experimental results indicate that the deep network transfer learning algorithm proposed in this article has certain application value for grape cluster quality grading.
- In summary, this study proposes a deep network transfer learning algorithm for grape cluster quality grading, which has achieved promising results on the red table grape dataset. The proposed method has the potential to enhance transparency in the fruit market, increase consumer trust, and ultimately support the development

and sustainability of the entire industry. However, the current study focuses on red grapes, utilizing specialized image acquisition equipment under specific indoor lighting conditions for grading research. There has been no investigation into the appearance quality grading of white grapes. The research on automated grading of grape clusters under natural light conditions remains a significant area of interest. In future research, our focus will be on developing an online grading model to assess the appearance quality of various categories of field grapes under natural light conditions. Additionally, we will work on the hardware deployment of the proposed model to ensure its practical applicability. This will be accomplished through the utilization of knowledge distillation or pruning algorithms. Furthermore, we aim to apply the model to field grading scenarios, allowing for practical implementation in real-world settings. For the unbalanced experimental sample in this study, we will focus on using image processing techniques, such as GAN, to solve the problem of balancing samples of different grades and to complete the construction and testing of a system for grading the quality of red grapes.

**Author Contributions:** Conceptualization, J.W.; Methodology, J.W., Z.Z. and D.S.; Software, Z.Z.; Validation, Z.Z.; Formal analysis, Z.Z.; Investigation, D.S., H.S. and X.C.; Resources, J.W. and Z.Z.; Data curation, Z.Z., X.C. and D.S.; Writing—original draft preparation, Z.Z. and D.S.; Writing—review and editing, Z.Z.; Manuscript revising, J.W. and H.S.; Study design, Z.Z., X.C. and J.W.; Supervision, J.W., H.S. and D.S.; Project administration, Z.Z. and J.W.; Funding acquisition, J.W. All authors have read and agreed to the published version of the manuscript.

**Funding:** This work was funded by the National Natural Science Foundation of China (grant number 31860466). The project is funded by Professor Jie Wu.

**Data Availability Statement:** Not applicable.

**Conflicts of Interest:** The authors declare no conflict of interest.

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
