# Peer review of "Classification of Appearance Quality of Red Grape Based on Transfer Learning of Convolution Neural Network"

_agronomy, doi:10.3390/agronomy13082015_

Round 1
Reviewer 1 Report
see the attachment

Moderate editing of English language
Author Response
Dear Editor and Reviewers,
Thank you for your letter and for the reviewers’ comments concerning our manuscript entitled “Classification of appearance quality of red grape based on transfer learning of convolution neural network” (Manuscript ID: agronomy-2425232). Those comments are all valuable and very helpful for revising and improving our paper, as well as the important guiding significance to our researches. We have studied the comments carefully and have made correction which we hope meet with approval. Revised portions are marked in underlined text in the paper. We have responded to every comment from Reviewers. Please see the attachment to our submission for details.
We have uploaded the code as well as part of the dataset already to GitHub at: https://github.com/Ayeenchante/Graded-code-files-MATLAB-.git
Yours sincerely,
Dongyuan Shi
ayeenchante@gmail.com

Reviewer 2 Report
1. Provide a comprehensive analysis of the limitations of manual grading methods for red grape external appearance, including subjectivity, inefficiency, and low accuracy.
2. Clearly define and explain the specific morphological or visual features that are crucial for accurate red grape grading, highlighting their significance in the grape industry.
3. Conduct a thorough literature review on existing CNN-based methodologies for fruit grading, identifying gaps and areas for improvement.
4. Describe the red grape dataset used, including its size, diversity, and any preprocessing techniques applied.
5. Discuss evaluation metrics used to quantify the performance of the proposed model, and compare it with state-of-the-art grading methods.
6. Analyze potential limitations and challenges of the proposed model, considering grape variety variations, lighting conditions, and image noise.
7. Provide a detailed analysis of factors contributing to the achieved accuracy rate, including performance across different grape varieties and hyperparameter tuning strategies.
8. Discuss scalability and feasibility of implementing the model in real-world grape industry settings, addressing computational requirements and deployment challenges.
9. Identify future research directions, such as exploring advanced CNN architectures or integrating multi-modal data sources, to enhance the accuracy and practicality of red grape grading systems.
10. Explain about the hyper parameters selection problem. Not mentioned
Difficult to understand some lines due to ambiguity in text.
Author Response

(The authors gave the same response as above.)

Round 2
Reviewer 1 Report
Author has incorporated all the comments
Minor editing of English language required
Author Response
Dear Editor and Reviewers,
?Thank you for your letter and for the reviewers’ comments concerning our manuscript entitled “Classification of appearance quality of red grape based on transfer learning of convolution neural network” (Manuscript ID: agronomy-2425232). Those comments are all valuable and very helpful for revising and improving our paper, as well as the important guiding significance to our research. We have studied the comments carefully and have made correction which we hope meet with approval. Revised portions are marked in underlined text in the paper. We have responded to every comment from Reviewers. Please see the attachment to our submission for details.
We have uploaded the code as well as part of the dataset already to GitHub at: https://github.com/Ayeenchante/Graded-code-files-MATLAB-.git
Yours sincerely,
Dongyuan Shi
ayeenchante@gmail.com
Reviewer 2 Report
->The introduction should provide more context and background information on the existing challenges in red grape grading and explain why traditional manual grading is inefficient.
-> Clarify the data enhancement techniques used to generate the dataset of 1800 samples. Explain how these enhancements preserve the authenticity of the grape images.
-> The selection of the five pre-trained CNN models (AlexNet, VGG16, GoogleNet, InceptionV3, and ResNet50) for transfer learning lacks justification. Provide reasoning for choosing these specific models and how they relate to the red grape grading task.
-> The fine-tuning process for each of the five CNN models should be explained in more detail. Include information on the specific layers fine-tuned, hyperparameters used, and convergence criteria.
-> Provide a comparison of the classification performance for each of the five CNN models without fine-tuning to assess the impact of fine-tuning on the results.
-> Explain why ResNet50 with a learning rate of 0.001 and a loop number of 10 was chosen as the best feature extractor. Include relevant metrics and comparisons with other models.
-> The analysis of different convolutional layer features output by the ResNet50 feature extractor requires more clarity. Elaborate on the methodology and how the analysis influenced the final model.
-> Provide more details about the SVM classifier, including kernel choice, hyperparameter tuning, and performance evaluation.
-> The term "the feature parameters extracted from the 10th node of the ResNet50 network" needs clarification. Specify which specific layer or node of the network is being referred to.
-> The experimental setup and evaluation metrics used for assessing the accuracy of the proposed model should be clearly described and augmented.
->Discuss the limitations of the proposed model, such as the dependence on specific camera equipment and the generalizability to different grape varieties or environmental conditions.
-> Address the ethical considerations related to data collection, especially when using vineyards for research purposes.
Still needs improvements, too many long and ambiguous lines.
Author Response

(The authors gave the same response as above.)

Round 3
Reviewer 2 Report
I have no more suggestions
I have no more suggestions